# Evaluation of renal injury and function biomarkers, including symmetric dimethylarginine (SDMA), in the rat passive Heymann nephritis (PHN) model

**Michael J. Coyne**  [1]*, **A. Eric Schultze**[2], **Donald J. McCrann, III**[1], **Rachel E. Murphy**[1], **Julie Cross**[1], **Marilyn Strong-Townsend**[1], **Corie Drake**[1], **Rebekah Mack**[1]

1 IDEXX Laboratories, Inc., Westbrook, Maine, United States of America, 2 Pathology Department, Toxicology Division, Lilly Research Laboratories, Eli Lilly and Company, Indianapolis, Indiana, United States of America

* michael-coyne@idexx.com

## Abstract

Symmetric dimethylarginine (SDMA) is a serum biomarker of excretory renal function which consistently correlates with glomerular filtration rate (GFR) across multiple species including rats, dogs, and humans. In human and veterinary clinical settings SDMA demonstrates enhanced sensitivity for detection of declining renal function as compared to other serum biomarkers, but application in preclinical study designs thus far has been limited. The purpose of this study was to determine the performance of serum SDMA in a rat passive Heyman nephritis model of glomerulopathy. In addition to SDMA other biomarkers of excretory renal function were measured including serum creatinine (sCr), blood urea nitrogen (BUN), and cystatin C along with creatinine clearance. Urinary renal biomarkers including microalbumin (μALB), clusterin (CLU), cystatin C, kidney injury marker-1 (KIM-1), neutrophil gelatinase-associated lipocalin (NGAL), and osteopontin (OPN) were also measured. PHN was induced using commercial sheep anti-Fx1A serum. Tissue, serum, and urine were collected from groups of control and anti-Fx1A-treated animals for biomarker evaluation, hematology, urinalysis, serum biochemistry, and histologic examination of kidney. Over the course of a 28-day study, concentrations of the urinary biomarkers μALB, CLU, cystatin C, NGAL, KIM-1 and the serum biomarker cystatin C increased significantly in anti-Fx1A-treated rats as compared to controls but no significant increase in serum SDMA, sCr, BUN, or creatinine clearance were noted in anti-Fx1A-treated rats. Given lack of direct GFR measurement or significant change in the renal function biomarkers sCr, BUN, and creatinine clearance, it is unclear if GFR differed significantly between control and anti-Fx1A-treated rats in this study, though urinary biomarkers and histopathologic findings supported renal injury in anti-Fx1A-treated rats over the time course investigated. This study is among the first to investigate serum SDMA in a rat model relevant to preclinical safety assessment and serves to inform future experimental designs and biomarker selection when evaluation of glomerular injury is of priority.

**Data Availability Statement:** All relevant data are within the paper and its Supporting Information files.

**Funding:** This work was funded by IDEXX Laboratories, Inc., Westbrook, Maine, USA.The funders had no role in study design, data collection and analysis, decision to publish, or preparation of the manuscript.

**Competing interests:** I have read the journal's policy and the authors of this manuscript have the following competing interests:Michael J. Coyne, Donald J. McCrann III, Rachel Murphy, Julie Cross, Marilyn Strong-Townsend, Corie Drake, and Rebekah M. Mack are current employees of IDEXX Laboratories, Inc. A. Eric Schultze is an employee of Eli Lilly and Company.

## Introduction

Renal biomarker discovery is an active and burgeoning area of research. Applications of a promising biomarker may be far ranging, including safety evaluation in the preclinical and clinical phases of drug development as well as clinical diagnosis and monitoring of renal disease. Suitability of a biomarker for these various assessments should be determined by studies designed to evaluate the intended use in the population or species of interest. To this end, numerous publications have evaluated emerging renal biomarkers in preclinical and clinical contexts, and in 2008, eight urinary nephrotoxicity biomarkers were the first to be qualified under the Food and Drug Administration's (FDA) biomarker qualification process [1–4]. As the relative merits of an individual renal biomarker may vary by application, emerging biomarkers can provide complimentary information when evaluated in concert. For instance, while the urinary nephrotoxicity biomarkers KIM-1 and β2-microglobulin were both qualified by the FDA as safety biomarkers to assess renal injury in the rat, KIM-1 is a biomarker for tubular alterations and β2-microglobulin is used to detect glomerular damage or impairment of kidney tubular reabsorption [4]. Biomarkers can vary in expression level, localization within the nephron, and response time-course following renal injury; as such assessment of renal biomarkers using a panel approach can deepen understanding of pathologic processes, and when measured longitudinally, enhance evaluation for improvement or decline in renal function both in preclinical models and clinical settings [5].

Symmetric dimethylarginine (SDMA) is a biomarker of excretory renal function. A byproduct of intranuclear arginine methylation, SDMA is produced in a stable manner by all nucleated cells, released to the serum as intracellular proteins are processed, and excreted primarily (>90%) by renal clearance [6]. In people, dogs, cats, and rats, serum SDMA has been shown to correlate highly to glomerular filtration rate (GFR) as estimated by inulin clearance or other surrogate markers such as creatinine clearance [6–9]. Though still routinely employed, the traditional serum biomarkers of renal excretory function creatinine and blood urea nitrogen (BUN) have long been criticized as relatively insensitive for early detection of declining renal function. The specificity of these biomarkers for renal function can also be complicated by comorbidities including loss of muscle mass in the case of creatinine, or gastrointestinal pathology in the case of BUN. As such, much research has focused on noninvasive serum biomarkers such as SDMA and cystatin C as more sensitive alternatives to traditional serum biomarkers of excretory renal function. In the veterinary clinical setting, serum SDMA was a more sensitive biomarker than serum creatinine for declining renal function in dogs and cats with naturally occurring chronic kidney disease (CKD), increasing with as little as 25% loss of excretory function [6, 9]. In pediatric human patients, serum SDMA showed higher diagnostic efficiency than serum cystatin C for detecting CKD [10]. As SDMA has been shown to outperform other surrogate serum biomarkers of GFR across multiple species including rats and dogs, it may be a promising candidate to enhance understanding of renal excretory function in preclinical studies utilizing these species.

Investigations of SDMA in the peer-reviewed literature predominantly focus either on utility as a diagnostic biomarker in naturally occurring renal disease or basic science research, while evaluations of SDMA as a renal safety biomarker in relevant preclinical species and study designs are not as well represented. Recent validation of a high through put immunoassay for measurement of serum SDMA in rats, however, allows for increased opportunities to investigate SDMA as a safety biomarker in this species[11]. The purpose of this study was to evaluate the utility of serum SDMA as a biomarker of renal excretory function within a rat passive Heymann nephritis (PHN) model of glomerulopathy. Other biomarkers of renal excretory function including serum creatinine, serum cystatin C, creatinine clearance, and the urinary

renal injury markers μALB, CLU, cystatin C, KIM-1, NGAL, and OPN were measured to evaluate how SDMA compliments biomarkers currently employed in preclinical toxicity study designs. Renal histopathologic examination was performed to confirm expected histologic findings of the PHN model were recapitulated in this design, and to compare biomarker data with light microscopic findings.

## Materials and methods

### Animals

Male Sprague Dawley CD® IGS rats (Charles River Laboratories, Raleigh, NC, USA), approximately 7–9 weeks old, weighing 150–350 g were used in the study conducted at Covance Laboratories Inc., (Greenfield, IN, USA). The animal facility where rats were group housed was limited access, with temperature and relative humidity maintained between 20 to 26°C, a relative humidity of 30 to 70%, and a 12-hour light/12-hour dark cycle. Rats were acclimatized for a minimum of 3 days prior to onset of treatment. Rats were supplied *ad libitum* with Greenfield city water and a certified rat diet (#2014C Envigo, RMS, Inc.) and were given various cage-enrichment devices and dietary enrichment on full feeding days.

### Facilities and animal use statement

Study protocol 8363519 was reviewed and approved in February 2017 by the Institutional Animal Care and Use Committee at Covance laboratories (Greenfield, IN). The facility used in this study was approved by the Association for Assessment and Accreditation of Laboratory Animal Care International, and the care and use of animals were in accordance with the Guide for the Care and Use of Laboratory Animals.

### Test material and treatment protocol

Glomerulonephritis was induced in the rat population following the PHN model using anti-Fx1A (commercial sheep anti-Rat Fx1A serum (PTX-002S), Probetex, Inc, (San Antonio, TX, USA). 0.9% sodium chloride for injection was used as vehicle control. PHN test article, dose, route of administration, and vehicle control selection were based upon previous recommendations for model development [12]. A preliminary study for determination of dose response and time course was performed to guide dose selection and verify onset of proteinuria. Study design for the preliminary dose determination phase and longitudinal phase are detailed in Table 1. Three anti-Fx1A serum doses (2.5 mL/kg, 5.0 mL/kg and 7.5 mL/kg) were evaluated in the dose determination phase in 36 rats and following selection of an optimal dose of anti-

**Table 1. Study design.**

| Treatment Group | Test Article | Dose (mL/kg) | Route | Pilot Dose Determination Phase | | | Longitudinal Phase | | | |
|---|---|---|---|---|---|---|---|---|---|---|
| | | | | Sample day after treatment (# of rats) | | | Sample day after treatment (# of rats) | | | |
| 1 | Vehicle | 0 | iv | 3 (3) | 9 (3) | 16 (3) | 9(12) | 16(12) | 21 (12) | 28 (12) |
| 2 | Anti-Fx1A | 2.5 | iv | 3 (3) | 9 (3) | 16 (3) | --- | --- | --- | --- |
| 3 | Anti-Fx1A | 5.0 | iv | 3 (3) | 9 (3) | 16 (3) | --- | --- | --- | --- |
| 4 | Anti-Fx1A | 7.5 | iv | 3 (3) | 9 (3) | 16 (3) | 9(12) | 16(12) | 21 (12) | 28 (12) |

---Denotes field not applicable, iv = intravenous

Fx1A serum (7.5 mL/kg), the study reported here-in was conducted in 96 rats. For both study phases, rats were randomly assigned to treatment group using a computerized procedure designed to achieve body weight balance with respect to group assignment. Rats were given slow bolus intravenous injection in the tail-vein, using a 25-gauge needle. The maximum volume of test article or vehicle injected was 7.5 mL/kg. Rats were dosed once.

## Body weight, clinical observations and mortality

Each rat was weighed at the time of allocation to treatment group and prior to necropsy. Rats were then observed twice daily to evaluate food and water intake, signs of pain or distress and for clinical signs of illness. Any change in food and water intake or clinical signs was recorded. Following injection, rats were observed for the first 4 hours for any for signs of distress or reaction to the injection.

Based upon documents that described anti-Fx1A-treated rats and pre-study dose ranging investigations completed prior to this study, no study-related pain, morbidity, or mortality was anticipated. The contract research organization's standard criteria for visible indicators of pain in rats on study included orbital tightening, nose/cheek flattening, changes in ear and whisker carriage, hunched posture, piloerection, and porphyrin staining around nose and muzzle [13, 14]. There were no specific euthanasia criteria outside of the institution's standard criteria. The standard criteria for euthanasia of animals on study included inability of animals to eat and drink, weight loss of $\geq$ 20% pre-study body weight, dehydration non-responsive to supportive care, and obvious signs of pain or suffering. Medical treatment necessary to prevent unacceptable pain and suffering, including euthanasia, was the sole responsibility of the attending laboratory animal veterinarian.

## Urine and blood collection

Rats were placed in individual metabolic cages and fasted the night prior to necropsy. Urine collected chilled over this 12–16 hour period was submitted to the clinical pathology laboratory for urinalysis, urine creatinine and urine biomarker measurement. On the day of necropsy, rats were anesthetized with isoflurane by inhalation and blood collected for routine clinical pathology and biomarker analysis. Blood for hematologic analysis and clinical chemistry determinations were obtained from the orbital plexus and transferred to $K_2EDTA$ and non-additive tubes, respectively. Blood for biomarker analysis was obtained from the abdominal aorta and transferred into non-additive tubes to harvest serum.

## Euthanasia and necropsy

After collection of blood for routine clinical pathology and biomarker analysis as described, rats were euthanized using isoflurane anesthesia and exsanguination, and a complete necropsy was performed.

## Clinical pathology and kidney biomarkers

Complete blood counts were obtained using an ADVIA 120 Hematology System with Multi-species Software (Version 3.1, Siemens Medical Solutions, Norwood, MA, USA) and Siemens reagents. Hematology parameters included: erythrocyte concentration, hematocrit (HCT), hemoglobin concentration, mean corpuscular hemoglobin (MCH), mean corpuscular hemoglobin concentration (MCHC), mean corpuscular volume (MCV), reticulocyte, total and differential leukocyte, and platelet counts. Differential leukocyte counts and blood cell

morphology were reviewed manually on Wright-Giemsa stained (ADVIA S60 Auto Slide Stainer, Siemens Medical Solutions) blood smears.

Clinical chemistry parameters were obtained using a Modular P Analyzer (Roche Diagnostics, Nutley, NJ) and Roche reagents. Measured serum values were obtained for the following: concentrations of creatinine, urea, sodium, chloride, potassium, inorganic phosphorous, calcium, albumin, total protein, cholesterol, triglycerides, glucose, total bilirubin, and activities of alanine aminotransferase (ALT), alkaline phosphatase (ALP), gamma-glutamyltransferase (GGT), aspartate aminotransferase (AST), and creatine kinase (CK). Calculated values were reported for globulins concentration and the albumin/globulin ratio.

The same methodology and reagents described above for serum creatinine (sCr) were used for measurement of urine creatinine (uCr). Values for uCr and sCr were used to determine creatinine clearance via previously reported methodology [15]. Creatinine clearance was adjusted for body weight and reported in units of mL/min/kg.

Complete urinalysis was performed using standard methods and consisted of the following parameters: urine color, clarity, volume, specific gravity, pH, protein, blood, ketones, glucose, bilirubin, urobilinogen, and sediment examination. Concentration of the urine biomarkers, µALB, CLU, cystatin C, KIM-1, NGAL, and OPN were measured and then normalized to urine creatinine concentration for reporting.

The urine cystatin C, µALB and NGAL were measured using the Luminex (Fl-labeled beads) platform (MILLIPLEX Rat Kidney Toxicity Magnetic Bead Panel 2 –Toxicity Multiplex Assay RKTX2MAG-37K, EMD Millipore Corporation, Billerica, MA USA) [16]. Biovendor ELISA method was used to obtain urine CLU (RD391034200CS) and serum cystatin C (RD391009200R) (BioVendor, Brno, Czech Republic) [17]. The urine KIM-1 and urine OPN were performed using R&D Systems ELISAs (RKM100 for KIM-1 and MOST00 for OPN) (R&D Systems, Inc. Minneapolis, MN USA) [18, 19]. Serum SDMA was measured using the IDEXX SDMA® Test (IDEXX Laboratories, Inc, Westbrook, ME, USA) [11].

## Anatomic pathology

Following gross examination of both kidneys, the right kidney was fixed in 10% neutral-buffered formalin for microscopic evaluation. Kidneys were embedded in paraffin, sectioned at 5 µm and stained with hematoxylin and eosin (H and E) or periodic acid-Schiff (PAS) stains. As the PHN model induces progressive glomerular injury with subsequent tubular injury, grading scheme for microscopic kidney alterations included assessment of glomerular and tubular changes on H and E- and PAS-stained sections [12]. Histopathologic evaluation was performed on the right kidney on all rats in all groups. Primary evaluation was performed by a board-certified veterinary pathologist and results were then peer-reviewed by an additional board-certified veterinary pathologist.

## Laboratories

Routine clinical pathology tests including clinical chemistry, hematology, urinalysis, and urine creatinine concentration were measured following standard methods at Covance Laboratories Inc. (Greenfield, IN). Anatomic pathology (gross and microscopic) was evaluated at Covance Laboratories Inc. (Greenfield, IN). Urine biomarkers for KIM-1, NGAL, urine Cystatin C and µALB were tested at Charles River Labs (CRL) (Mattawan, MI). Urine biomarker levels for CLU and OPN were measured at Eli Lilly and Company, (Indianapolis, IN). Serum Cystatin C was measured at Covance Laboratories Inc. (Greenfield, IN). SDMA was measured at IDEXX Laboratories, Inc. (Westbrook, ME).

## Statistical analysis

For all continuous variables subjected to statistical analysis Tukey's fence method was used to screen for extreme points within each timepoint and treatment group. Any point that was more than 4 times the inter quartile range below or above the first or third quartiles respectively was excluded from the analysis. To better approximate a normal distribution within groups, results were subjected to a log transformation before analysis. 2-Sample t-tests were used to compare treatment to a control group at each timepoint. Variance was not assumed to be equal between groups. For ordinal variables, Mann-Whitney U tests were used to identify significant differences between treatment and control groups at each timepoint. Significance was determined as $p \leq 0.05$ for each test without adjustment for multiple comparisons. All analyses were conducted in R version 4.0.0 [20]. For complete blood count, clinical chemistry and body weight data statistically significant differences reported in results were further defined by percent change from vehicle control at a given timepoint as follows: [(mean value $_{control\ group}$—mean value $_{treatment\ group}$)/ mean value $_{control\ group}$] X 100.

# Results

## Body weight, clinical observations and mortality

No mortality was observed in any of the rats during treatment. Most anti-Fx1A-treated rats had changes to the skin of the feet and ears (red discoloration) and one rat had labored breathing. All clinical signs were observed following dosing on day 1 and resolved after 1 hour of observation. All control and treatment groups demonstrated weight gain over the course of the study (S1 Table in S1 File). An 8%decrease in body weight at day 28 was noted in anti-Fx1A-treated animals as compared to vehicle controls (S1 Table in S1 File).

## Kidney biomarkers

Serum and urinary kidney biomarker data from vehicle control and anti-Fx1A-treated rats are presented in Table 2 and Figs 1–3. Two measures were identified as outliers and excluded from the analysis (NGAL, 0 ng/mg, Day 28 treatment group; OPN, 134.3 ng/mg, Day 9 treatment group). SDMA concentrations were not significantly different in anti-Fx1A-treated as compared to control rats for the duration of the study while serum Cystatin C concentrations were significantly increased ($P \leq 0.05$) in anti-Fx1A-treated rats at sampling days 9–21, but were not significantly different on day 28 (Fig 1, Table 2). sCr concentrations were not significantly different ($P \leq 0.05$) in anti-Fx1A-treated as compared to control rats. There was no significant difference in creatinine clearance in anti-Fx1A-treated rats as compared to controls (Fig 2, Table 2).

In anti-Fx1A-treated rats there were significant increases ($P \leq 0.05$) with various time courses for the urine biomarkers as compared to the controls (Table 2, Fig 3). These included an increase in urine µALB, cystatin C, and NGAL concentrations on days 9, 16, 21 and 28. Increases in urine CLU and KIM-1 concentrations were also present on days 16 and 21, and CLU remained increased while KIM-1 was not significantly different between anti-Fx1A-treated rats and controls on day 28. There was a significant decrease ($P \leq 0.05$) in urine OPN in anti-Fx1A-treated rats on days 9, 16, 21, and 28 as compared to vehicle controls.

## Clinical pathology

On complete blood count, multiple parameters were significantly decreased in anti-Fx1A-treated rats as compared to vehicle controls at multiple time points (S2 Table in S1 File). These included a decrease in HCT on days 9, 16, and 21 (4%) and on day 28 (8%), a decrease in

**Table 2. Serum and urine kidney biomarker data for vehicle control and anti-Fx1A-treated rats.**

| Treatment | Vehicle | | | | Anti-Fx1A 7.5 mL/kg | | | |
|---|---|---|---|---|---|---|---|---|
| Sample Day Post-treatment | Day 9 | Day 16 | Day 21 | Day 28 | Day 9 | Day 16 | Day 21 | Day 28 |
| Creatinine Clearance (mL/min/kg) | 3.77 (±0.67) | 4.06 (±0.70) | 4.16 (±0.77) | 3.66 (±0.84) | 3.51 (±0.83) | 3.97 (±0.72) | 3.76 (±0.93) | 4.01 (±0.87) |
| Creatinine (mg/dL) serum | 0.59 (±0.03) | 0.62 (± 0.06) | 0.62 (±0.06) | 0.62 (±0.04) | 0.58 (± 0.04) | 0.60 (±0.04) | 0.58 (±0.06) | 0.59 (±0.05) |
| Urea nitrogen (mg/dL) | 9.1 (± 1.4) | 10.3 (± 2.1) | 11.0 (± 2.1) | 12.7 (± 1.7) | 9.6 (± 1.3) | 11.1 (± 1.8) | 8.6* (± 1.4) | 9.9* (± 1.8) |
| Cystatin C (ng/mL) serum | 1773.9 (±290.8) | 1833.7 (±319.7) | 1558.1 (±309.9) | 1713.5 (±382.7) | 2274.9* (±450.3) | 2226.3* (±403.2) | 2178.3* (±363.0) | 1852.8 (±642.9) |
| SDMA (µg/dL) serum | 8.7 (±1.0) | 9.8 (±1.2) | 8.3 (±1.5) | 7.3 (±1.5) | 9.2 (±1.7) | 10.1 (±1.0) | 9.4 (±1.8) | 8.0 (±1.9) |
| µAlbumin (µg/mg) urine | 72.1 (±49.6) | 64 (±36.6) | 48.4 (±16.4) | 33.5 (±8.4) | 25782.8* (±13378.2) | 106776.5* (±70498.4) | 110247* (±34997.9) | 104792.8* (±43163.6) |
| Clusterin (ng/mg) urine | 87.3 (±74.5) | 67.9 (±269.5) | 84 (±188.2) | 47.5 (±469.8) | 113.2 (±48.5) | 275.4* (±25.7) | 305.3* (±31.2) | 449.5* (±20.1) |
| Cystatin C (ng/mg) urine | 1044.6 (±237.9) | 981 (±331.3) | 797.1 (±198.8) | 774.3 (±130.3) | 3647.6* (±3604.9) | 2898.3* (±1664.4) | 2748.3* (±879.7) | 2506.4* (±1068.6) |
| KIM-1 (pg/mg) urine | 432.4 (±261.1) | 512.7 (±206.4) | 357.8 (±267.8) | 514.2 (±140.2) | 773.7 (±408.2) | 1112.5* (±650.5) | 829.0* (±336.2) | 1367.6 (±1206.8) |
| NGAL (ng/mg) urine | 310.4 (±102.4) | 267.9 (±104.8) | 267.5 (±56.8) | 311.5 (±111.7) | 971.5* (±626.7) | 959.2* (±491.7) | 784.9* (±321.0) | 936.3* (±666.2) |
| Osteopontin (ng/mg) urine | 15.1 (±8.5) | 11.6 (±5.3) | 10 (±3.8) | 15.9 (±6.5) | 19.6* (±36.4) | 4.1* (±2.7) | 2.9* (±1.8) | 3.9* (±2.4) |

Data are presented as Mean (± SD)

[a] Values normalized to urine creatinine.

*Anti-Fx1A-treated group is significantly different (P ≤ 0.05) from vehicle control group at the same time point.

SDMA = symmetric dimethylarginine

NGAL = neutrophil gelatinase-associated lipocalin

KIM-1 = kidney injury marker-1

erythrocyte count on day 28 (7%), a decrease in hemoglobin concentration on day 28 (8%), a decrease in MCH on day 21 (2.7%), and a decrease in MCV on day 21 (2.7%). Other parameters were significantly increased in anti-Fx1A-treated rats as compared to vehicle controls at multiple time points (S2 Table in S1 File). These included an increase in MCHC on day 9 (1.9%), an increase in reticulocyte count on day 9 (26%) and day 28 (30%), an increase in absolute neutrophil counts on day 16 (40%) and day 28 (36%), and an increase in platelet count on day 9 (35%), day 16 (34%), day 21 (29%), and day 28 (28%).

On clinical chemistry evaluation, multiple analytes were significantly decreased in anti-Fx1A-treated rats as compared to vehicle controls at multiple time points (S3 Table in S1 File). These included a decrease in albumin concentration on day 9 (12%), and son day 16 (21%), day 21 (22%) and day 28 (24%); a decreased A:G ratio on day 9 (14%), and on day 21 (38%), day 21 (41%), and day 28 (40%); a decrease in total protein concentration on day 9 (5%), day 21 (3%), and day 28 (6%); a decrease in ALP activity on day 21 (14%), and day 28 (19%); a decrease in AST activity on day 21 (17%), and day 28 (19%); a decrease in total calcium concentration on day 9 (2%), and day 28 (2%); a decrease in BUN concentration on day 21 (22%), and day 28 (22%), and a decrease on glucose concentration on day 28 (11%). Other parameters were significantly increased in anti-Fx1A-treated rats as compared to vehicle controls at multiple time points (S3 Table in S1 File). These included an increase in cholesterol concentration on day 9 (19%), and on day 16 (146%), day 21 (193%), and day 28 (158%); an increase in triglyceride concentration on day 9 (50%) and on day 16 (192%), day 21 (153%), and day 28 (188%); an increase in globulin concentration on day 16 (26%), day 21 (37%), and day 28 (29%), and an increase in potassium concentration on day 16 (11%) and day 21 (12%).

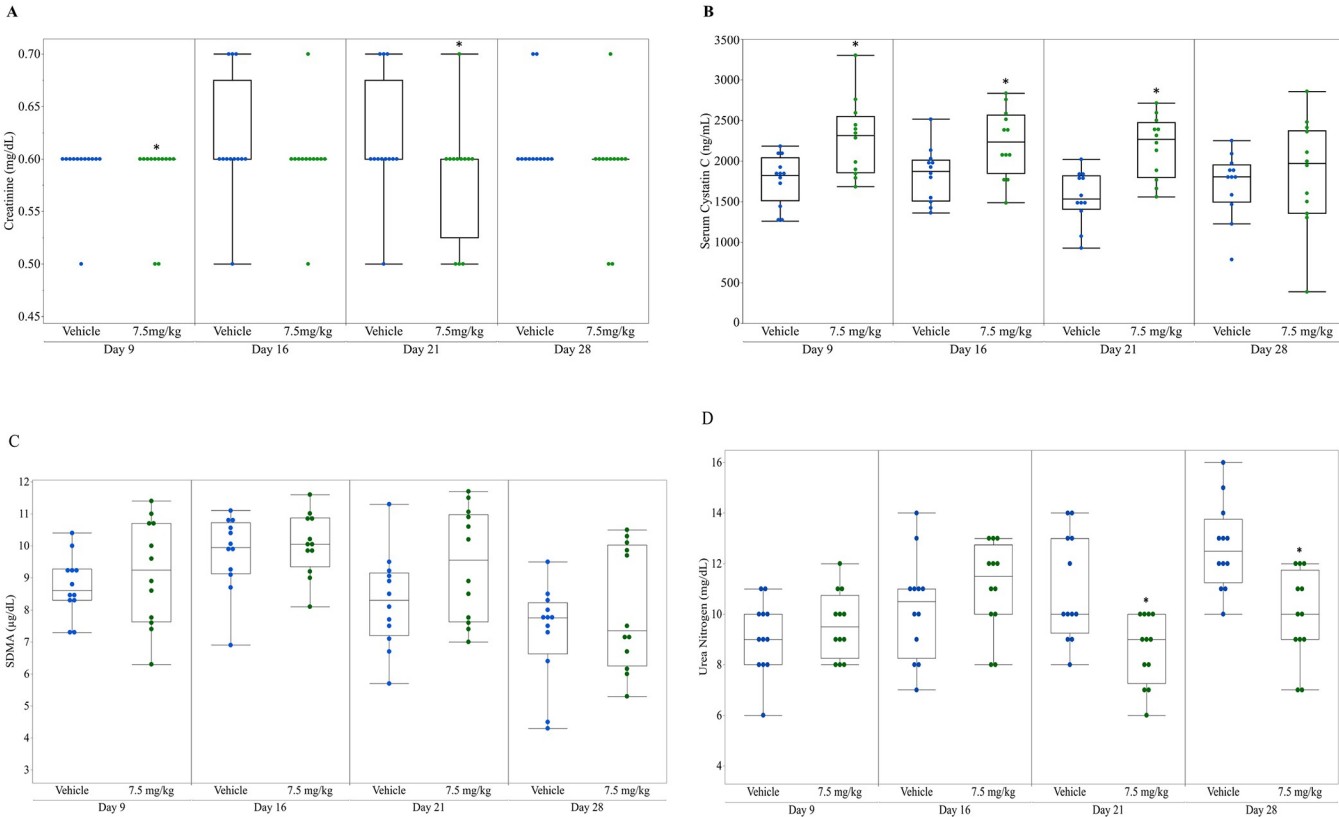

**Fig 1. Serum excretory renal function biomarkers in control and anti-Fx1A-treated rats.** A. Creatinine (mg/dL); B. Cystatin C (ng/mL); C. SDMA (μg/dL); D. Urea nitrogen (mg/dL). Outlier boxplot: Horizontal line within the box represents the median sample value, box represents interquartile range (IQR), whiskers extend to 1.5x IQR. * Indicates treatment group is significantly different (p ≤ 0.05) from vehicle control group at the same number of doses.

On urinalysis only urine protein was significantly increased in magnitude and frequency at all time points in anti-Fx1A-treated rats (P < 0.001) (S4 Table in S1 File).

## Anatomic pathology

The changes in microscopic appearance of the kidney in anti-Fx1A-treaded rats are listed in Table 3. Microscopic changes in the kidney were time-dependent. In anti-Fx1A-treated rats, increased mesangial matrix and basophilic tubules, interpreted as regeneration, were present on days 9, 16, 21, and 28. Dilation, degeneration and necrosis of renal tubules, proteinaceous casts, and infiltration of mixed population of inflammatory cells (mononuclear and neutrophils) were observed in rats on days 16, 21, and 28. Glomeruli contained expanded PAS-negative mesangial matrix and glomerular capillaries were compressed. Basophilic tubules were observed near degenerated tubules. Degeneration and necrosis were identified by one or more of the following features: hypereosinophilia, vacuolation, fragmentation of cytoplasm, pyknotic nuclei, and/or sloughed epithelial cells within tubular lumens. Tubular epithelium degeneration and necrosis were observed in the proximal and distal tubules, as well as collecting ducts, concentrated primarily in the cortex, and extended into the medulla and papilla. Dilated tubules were lined by flattened epithelial cells and contained degenerated cell debris or PAS-positive hyaline proteinaceous material. Hyaline droplets occurred in degenerated epithelial cells in anti-Fx1A-treated rats at day 16. Minimal infiltration of mononuclear cells was noted anti-Fx1A-treated rats on day 9 (Table 3).

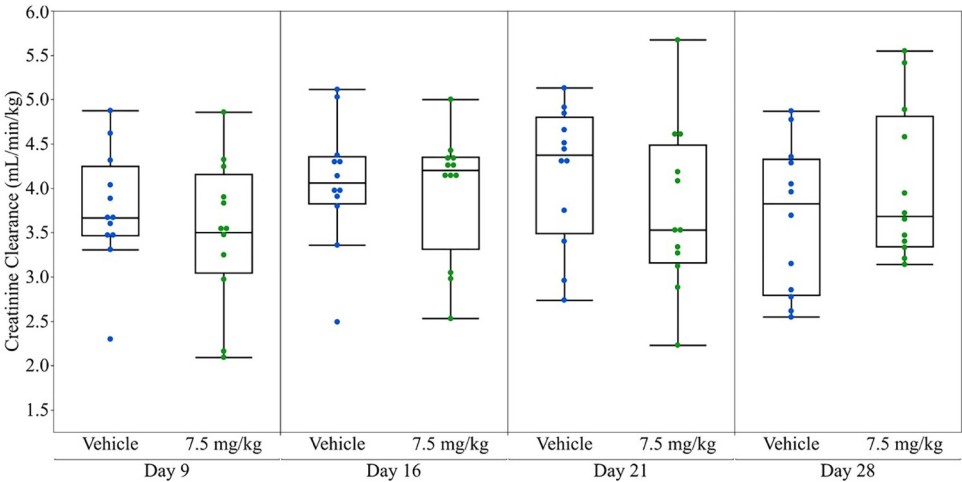

**Fig 2. Creatinine clearance (mL/min/kg) in control and anti-Fx1A-treated rats.** Outlier boxplot: Horizontal line within the box represents the median sample value, box represents interquartile range (IQR), whiskers extend to 1.5x IQR.

## Discussion

In preclinical studies immune-mediated glomerular damage represents the most commonly encountered form of drug or chemical-related glomerular injury [21]. In rats, PHN is a well-established model of membranous nephropathy, a leading cause of nephrotic syndrome in humans, and has been shown to recapitulate certain features of the human condition including hypoalbuminemia, hypercholesterolemia, and hypertriglyceridemia, as noted in anti-Fx1A-treated rats in this study [12, 22]. Clinical pathology, histopathologic assessment, as well as onset and degree of proteinuria in anti-Fx1A-treated rats were consistent with previous cases of immune-mediated glomerulopathy [12, 23, 24]. By light microscopy glomerular changes were minimal in anti-Fx1A-treated rats over the time course investigated, consisting of slight to mild proliferation of mesangial matrix. Results were not unexpected given that podocyte injury with marked proteinuria can exist without appreciable histologic abnormalities, and others have documented minimal glomerular changes on light microscopy within 4 weeks post-treatment in the rat PHN model [24, 25]. Histopathologic evidence of tubular injury has been previously described within PHN and can result from deposition of anti-Fx1A antibodies on the brush border of proximal tubular cells which is enriched in the target antigens megalin and RAP [12, 23, 26]. Proteinuria may have been another inciting factor, as injury was not confined to the proximal renal tubules in this study. Proteinuria has been linked to increased intrarenal complement activation and tubular cell apoptosis, among other mechanisms which may drive tubular injury in glomerulopathies [27, 28]. Presence of mild mixed inflammatory cells in the kidney of anti-Fx1A-treated animals in this study coincided with observance of tubular degeneration and necrosis and may have been secondary to the observed tissue damage. In one study renal interstitial inflammation was abrogated in PHN rats treated with an ACE inhibitor, indicating proteinuria may also be a driver of inflammation in this model [29]. For anti-Fx1A-treated animals in this study, proteinuria as evidenced by urinalysis and increased μALB concentration was noted at initial assessment on day 9, persisted through day 28. Detection of proteinuria by soluble biomarkers preceded light microscopic findings of proteinaceous tubular casts observed in anti-Fx1A-treated animals beginning on day 16. These results support the conclusion of others that urinary biomarkers of proteinuria, such as urinary protein or albumin concentration, are among the most useful analytes for detection of

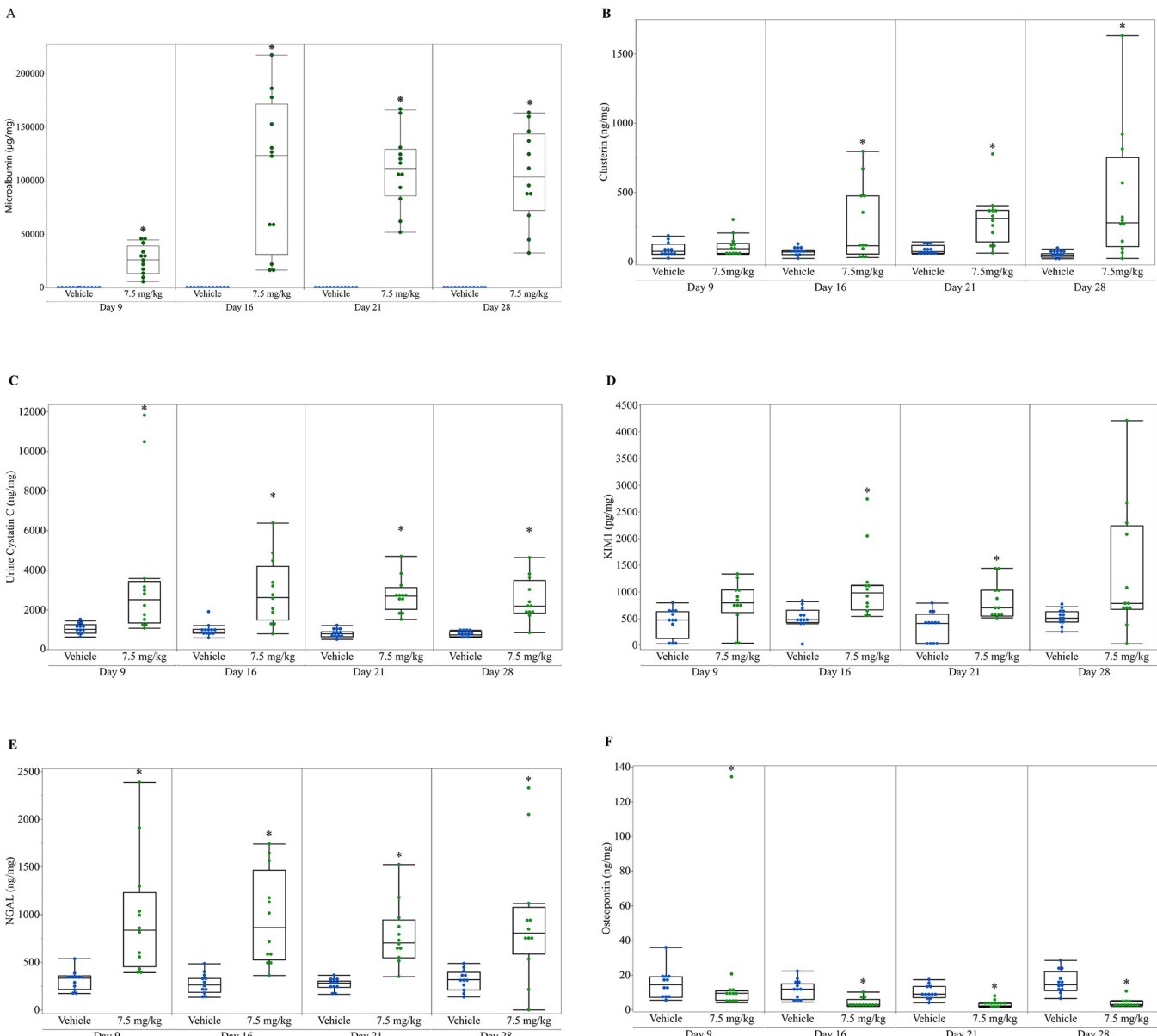

**Fig 3. Urine renal biomarkers in control and anti-Fx1A-treated rats.** A. Microalbumin (μg/mg); B. Clusterin (ng/mg); C. Cystatin C (ng/mg); D. KIM-1 (pg/mg); E. NGAL (ng/mg); F. Osteopontin (ng/mg). Urine biomarker values were normalized to urine creatinine concentration. Horizontal line within the box represents the median sample value, box represents interquartile range (IQR), whiskers extend to 1.5x IQR. * Indicates treatment group is significantly different (p ≤ 0.05) from vehicle control group at the same number of doses.

glomerulonephritis [21, 30]. The urinary biomarkers measured in this study have been previously qualified by the FDA for assessment of renal toxicity in rats and information on the function, localization, and relative performance of these biomarkers have been reviewed in a recent publication [4, 30].

In addition to μALB concentration, concentrations of the urinary biomarkers cystatin C and NGAL were increased, beginning on day 9 and persisting through day 28. As cystatin C and NGAL are both reabsorbed from the glomerular filtrate by the proximal tubules, these changes could reflect tubular dysfunction secondary to urinary protein overload; alternatively,

**Table 3. Kidney microscopic alterations in anti-Fx1A-treated rats.**

| Dose | Anti-Fx1A 7.5 mL/kg | | | |
|---|---|---|---|---|
| Time | Day 9 | Day 16 | Day 21 | Day 28 |
| Number of male rats | 12 | 12 | 12 | 12 |
| Microscopic alteration | Number of rats affected, severity score | | | |
| Mesangial matrix | 3 SL | 5 MI | 1 MI | 2 MI |
| | | 5 SL | 11 SL | 8 SL |
| Degeneration/necrosis, tubules | – | 5 MI | 2 MI | 4 MI |
| | | 4 MI | 6 SL | 4 SL |
| | | 3 MO | 3 MO | 3 MO |
| | | | | 1 MA |
| Proteinaceous casts | – | 2 MI | 2 MI | 4 MI |
| | | 5 SL | 8 SL | 6 SL |
| | | 2 MO | 1 MO | 1 MO |
| Dilated tubules | – | 2 MI | 2 MI | 3 MI |
| | | 4 SL | 8 SL | 4 SL |
| | | 3 MO | 1 MO | 3 MO |
| | | | | 1 MA |
| Basophilic tubules | 4 MI | 5 MI | 3 MI | 4 MI |
| | 1 SL | 5 SL | 8 SL | 4 SL |
| | | 1 MO | 1 MO | 4 MO |
| Infiltrate, mixed cells | – | 8 MI | 8 MI | 7 MI |
| | | 4 SL | 4 SL | 5 SL |
| Infiltrate, mononuclear cells | 10 MI | | – | – |
| Hyaline droplet, tubule cell | – | 1 MI | – | – |
| | | 4 SL | | |

Severity grading scale: — = Equivocal change or finding not observed; MI = Minimal; SL = Slight; MO = Moderate; MA = Marked

as reabsorption of both cystatin C and NGAL appears megalin-dependent, binding of Fx1A antibodies to megalin within the proximal tubular brush border may have contributed [31, 32]. Increased urinary NGAL concentration in anti-Fx1A-treated rats may also reflect tubular injury, as NGAL expression and secretion by renal tubular epithelial cells has been shown to increase with stimuli including inflammation or ischemic injury [33, 34]. KIM-1, a type 1 cell membrane glycoprotein found within proximal renal tubular epithelium, is upregulated and shed into the urine with proximal tubular injury in rats, likely accounting for the increase in urinary KIM-1 seen on day 16 and 21 in anti-Fx1A-treated rats [35, 36]. CLU, a secreted protein which originates from multiple segments of the renal tubule as well as mesangial cells, increased in urine on days 16, 21, and 28 in anti-Fx1A-treated animals, consistent with previous rat models of glomerulopathy [30, 37]. Significantly lower urine OPN in anti-Fx1A-treated rats beginning day 9 and persisting through day 28 was a somewhat unexpected finding. OPN is a secreted glycoprotein normally found in the loop of Henle and distal nephron and can be expressed by all tubule segments and the glomerulus following renal injury. Renal OPN expression was increased in renal tubular cells in PHN rats as well as in humans with naturally occurring glomerulopathies [27, 38]. In multiple animal models and human studies of glomerular disease, urinary OPN is unchanged or increased in xenobiotic- treated animals or affected individuals as compared to controls. In patients with IgA nephritis, however, urinary OPN was significantly decreased as compared with normal controls, despite upregulation of OPN

expression within renal epithelium [30, 39]. Subsequent immunoblot analysis identified a 34kD fragment of OPN in the urine of patients with IgA nephritis and in some patients with other glomerulopathies, and that this fragment could be induced in urine from normal controls on treatment with thrombin. These data suggest that decreased urinary OPN could reflect altered secretion or processing of OPN in glomerulopathy [39]. It is possible a similar mechanism may explain decreased urinary OPN in anti-Fx1A-treated rats in this study, though renal OPN expression and immunoblot analysis were not undertaken to further investigate this finding.

A primary goal of this study was to evaluate utility of serum SDMA measurement within a preclinical glomerular toxicity model. While urine biomarkers can identify and in certain instances localize renal injury, markers of excretory function are best suited to inform potential impact on GFR and global renal excretory function. Performance of SDMA was compared to other renal excretory function biomarkers creatinine clearance, sCr, BUN, and serum cystatin C. Serum SDMA, sCr, BUN, and creatinine clearance did not indicate a decline in excretory renal function and were not significantly increased in anti-Fx1A-treated rats as compared to controls. Results for creatinine clearance were consistent with previous findings that decline in this biomarker did not develop until >150 days on study and an investigation in which renal blood flow and creatinine clearance were found to be significantly different between anti-Fx1A-treated rats and controls in chronic (18 months) but not subacute (2 months) PHN [23, 40]. In contrast serum cystatin C was significantly increased in anti-Fx1A-treated rats on days 9, 16, and 21 but not day 28. Cystatin C, a protease inhibitor, shares certain physiologic similarities with SDMA including synthesis by all nucleated cells, renal excretion, and correlation with GFR [9, 41]. Serum cystatin C has been proposed as a more sensitive and specific biomarker for renal function than sCr, and in some studies has been shown to outperform sCr or creatinine clearance in estimation of GFR[42–44]. Independent of GFR, however, certain extra-renal factors in people have also been described to increase serum cystatin C, including hyperthyroidism and glucocorticoid administration [41, 45, 46]. Dexamethasone administration in rats has also been shown to increase plasma cystatin C levels without change in GFR, as measured by inulin clearance [47]. Therefore, it remains unclear in the current study if increased serum cystatin C in anti-Fx1A-treated rats reflects enhanced sensitivity for detection of decreased GFR as compared to SDMA, sCr, and creatinine clearance, or if other causes, such as extrarenal factors in anti-Fx1A-treated rats, may have contributed. Direct measure of GFR, such as by inulin clearance, may have helped to elucidate the most likely cause for increased serum cystatin C and lack of direct GFR measurement is considered a limitation of this study.

In conclusion, this study characterized the performance of various urinary and serum renal biomarkers within the PHN rat model. With the exception of OPN, urinary biomarkers performed as expected based upon results in other renal toxicity models, and clinical and anatomic pathology findings aligned with the previously well-described pathophysiology of PHN. Relative performance of SDMA as compared to other excretory renal function biomarkers was not able to be fully assessed in this model, as it remains unclear if GFR differed significantly between control and treated rats over the time course investigated.

## Supporting information

**S1 File.**
(DOCX)

**S1 Data.**
(XLSX)

## Acknowledgments

The authors wish to thank Diane Hamlin (Lilly) for excellent technical assistance in completion of this study and in biomarker analysis and Dr. Mary K Leissinger (IDEXX) for technical and editorial input for this project.

## Author Contributions

**Conceptualization:** Michael J. Coyne, A. Eric Schultze, Rebekah Mack.

**Data curation:** Rachel E. Murphy.

**Formal analysis:** Michael J. Coyne, Donald J. McCrann, III, Corie Drake.

**Investigation:** Julie Cross, Marilyn Strong-Townsend.

**Visualization:** Michael J. Coyne, Corie Drake.

**Writing – original draft:** Michael J. Coyne, A. Eric Schultze, Rebekah Mack.

**Writing – review & editing:** Michael J. Coyne, A. Eric Schultze, Donald J. McCrann, III, Rachel E. Murphy, Julie Cross, Marilyn Strong-Townsend, Corie Drake, Rebekah Mack.

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
