## [Decision Letter · Decision Letter 0]

28 Mar 2022

PONE-D-22-03958Evaluation of renal injury and function biomarkers, including symmetric dimethylarginine (SDMA), in the rat passive Heymann nephritis (PHN) modelPLOS ONE

Dear Dr. Michael J Coyne,

Thank you for submitting your manuscript to PLOS ONE. After careful consideration, we feel that it has merit but does not fully meet PLOS ONE’s publication criteria as it currently stands. Therefore, we invite you to submit a revised version of the manuscript that addresses the points raised during the review process.

Reviewer #1: The investigations of SDMA in the peer-reviewed literature predominantly focus on its use as a diagnostic biomarker rather than on its evaluation as a renal safety marker which is relevant for preclinical/toxicologic studies. However there are some major limitations to this study.

Major comments:

The main aim of this study was to assess the utility of SDMA within a preclinical glomerular toxicity model (Lines 91-93 and Line 424). To do so, the authors should have included direct GFR measurement. Indeed, sCr, BUN and creatinine clearance are very insensitive markers of excretory renal function and therefore negative results were to be anticipated. The authors can not conclude if there was any difference in GFR between control and anti-Fx1A-treated rats which implies no thorough conclusions can be drawn regarding the utility of SDMA (and of Cys C) in this experimental context.

The manuscript contains expectable findings (and already published) for the urinary biomarkers. These biomarkers have already long been qualified by the FDA to evaluate renal toxicity and function. The newer investigation is the SDMA measurement in this context. Therefore it was important to include direct GFR measurements to provide the possibility for a thorough interpretation of SDMA and of Cys C findings (including relative performance to other biomarkers).

Other comments:

Lines 21-23. The authors state that SDMA correlates highly with GFR across multiple species including rats, dogs and humans. Line 69-71 this is repeated including cats. Based on current literature, this statement should be nuanced. Indeed, in dogs, Pelander et al. (2019) showed that overall diagnostic performance of SDMA as a marker of impaired GFR was the same as that of creatinine (both R2 = 0.62, P < .001) and this result was in line with McKenna et al. (2019) who concluded serum creatinine and SDMA were both only moderately correlated with the renal clearance (R2 = 0.52 and 0.27 respectively, p < .001). In cats, Brans et al. (2020) showed that the correlation between SDMA and GFR (τ B = −0.57; P < .001) was moderate and the correlation between sCr and GFR was of the same magnitude (τ B = −0.56; P < .001). Braff et al. (2014) equally observed that the relationship between SDMA and GFR (R2 = 0.82, P < .001) was very similar to the relationship between serum creatinine and GFR (R2 = 0.81, P <.001). Also in humans, a large meta-analysis of Kielstein et al. (2006) could not demonstrate outstanding correlation of 1/SDMA with GFR (r = 0.77 - 0.85, P < .001).

Lines 139-150. Did the laboratory animal veterinarian use a scoring system for assessment of pain and well-being of the rats ? If yes, please state so.

Lines 193-197. Please provide for all assays used: either appropriate references that include validation data or include validation data.

Lines 200-208. Especially if animals are sacrificed, the maximum information should be gathered from the samples/study. It is not unexpected that glomerular changes on light microscopy were minimal in the antiFx1A treated rats. It is unclear why ME or IF techniques were not used. This would have allowed to bring the interpretation of the renal lesions to another level.

In the results section, the authors make too much subjective interpretation of their data (the terms minimal, slightly … are used numerous numerous times).

I don’t see a good argument why statistics were not performed for proteinuria ?

Discussion: the first paragraph is too long (Lines 348-390).

Line 438-440: please provide references for this statement on cystatin C.

We look forward to receiving your revised manuscript.

Kind regards,

Prabhakar Orsu, PhD

Academic Editor

PLOS ONE

Journal Requirements:

Reviewers' comments:

Reviewer's Responses to Questions

**Comments to the Author**

1. Is the manuscript technically sound, and do the data support the conclusions?

Reviewer #1: Partly

2. Has the statistical analysis been performed appropriately and rigorously? 

Reviewer #1: Yes

3. Have the authors made all data underlying the findings in their manuscript fully available?

Reviewer #1: Yes

4. Is the manuscript presented in an intelligible fashion and written in standard English?

Reviewer #1: Yes

5. Review Comments to the Author

Reviewer #1: The investigations of SDMA in the peer-reviewed literature predominantly focus on its use as a diagnostic biomarker rather than on its evaluation as a renal safety marker which is relevant for preclinical/toxicologic studies. However there are some major limitations to this study.

Major comments:

The main aim of this study was to assess the utility of SDMA within a preclinical glomerular toxicity model (Lines 91-93 and Line 424). To do so, the authors should have included direct GFR measurement. Indeed, sCr, BUN and creatinine clearance are very insensitive markers of excretory renal function and therefore negative results were to be anticipated. The authors can not conclude if there was any difference in GFR between control and anti-Fx1A-treated rats which implies no thorough conclusions can be drawn regarding the utility of SDMA (and of Cys C) in this experimental context.

The manuscript contains expectable findings (and already published) for the urinary biomarkers. These biomarkers have already long been qualified by the FDA to evaluate renal toxicity and function. The newer investigation is the SDMA measurement in this context. Therefore it was important to include direct GFR measurements to provide the possibility for a thorough interpretation of SDMA and of Cys C findings (including relative performance to other biomarkers).

Other comments:

Lines 21-23. The authors state that SDMA correlates highly with GFR across multiple species including rats, dogs and humans. Line 69-71 this is repeated including cats. Based on current literature, this statement should be nuanced. Indeed, in dogs, Pelander et al. (2019) showed that overall diagnostic performance of SDMA as a marker of impaired GFR was the same as that of creatinine (both R2 = 0.62, P < .001) and this result was in line with McKenna et al. (2019) who concluded serum creatinine and SDMA were both only moderately correlated with the renal clearance (R2 = 0.52 and 0.27 respectively, p < .001). In cats, Brans et al. (2020) showed that the correlation between SDMA and GFR (τ B = −0.57; P < .001) was moderate and the correlation between sCr and GFR was of the same magnitude (τ B = −0.56; P < .001). Braff et al. (2014) equally observed that the relationship between SDMA and GFR (R2 = 0.82, P < .001) was very similar to the relationship between serum creatinine and GFR (R2 = 0.81, P <.001). Also in humans, a large meta-analysis of Kielstein et al. (2006) could not demonstrate outstanding correlation of 1/SDMA with GFR (r = 0.77 - 0.85, P < .001).

Lines 139-150. Did the laboratory animal veterinarian use a scoring system for assessment of pain and well-being of the rats ? If yes, please state so.

Lines 193-197. Please provide for all assays used: either appropriate references that include validation data or include validation data.

Lines 200-208. Especially if animals are sacrificed, the maximum information should be gathered from the samples/study. It is not unexpected that glomerular changes on light microscopy were minimal in the antiFx1A treated rats. It is unclear why ME or IF techniques were not used. This would have allowed to bring the interpretation of the renal lesions to another level.

In the results section, the authors make too much subjective interpretation of their data (the terms minimal, slightly … are used numerous numerous times).

I don’t see a good argument why statistics were not performed for proteinuria ?

Discussion: the first paragraph is too long (Lines 348-390).

Line 438-440: please provide references for this statement on cystatin C.

6. PLOS authors have the option to publish the peer review history of their article (what does this mean?). If published, this will include your full peer review and any attached files.

Reviewer #1: No

---

## [Author Response · Author response to Decision Letter 0]

6 May 2022

The authors would like to thank the reviewer for their comments. We appreciate the time and effort in reviewing our manuscript. Our responses are below.

Reviewer #1: 

Major comments:

The main aim of this study was to assess the utility of SDMA within a preclinical glomerular toxicity model (Lines 91-93 and Line 424). To do so, the authors should have included direct GFR measurement. Indeed, sCr, BUN and creatinine clearance are very insensitive markers of excretory renal function and therefore negative results were to be anticipated. The authors can not conclude if there was any difference in GFR between control and anti-Fx1A-treated rats which implies no thorough conclusions can be drawn regarding the utility of SDMA (and of Cys C) in this experimental context.

The manuscript contains expectable findings (and already published) for the urinary biomarkers. These biomarkers have already long been qualified by the FDA to evaluate renal toxicity and function. The newer investigation is the SDMA measurement in this context. Therefore it was important to include direct GFR measurements to provide the possibility for a thorough interpretation of SDMA and of Cys C findings (including relative performance to other biomarkers).

The authors agree that direct measurement of GFR would have been optimal in this study, but the primary goal of this study was to evaluate utility of serum SDMA measurement within a well-defined preclinical glomerular toxicity model that uses creatinine clearance as a measure of renal function. Serum SDMA, sCr, BUN, and creatinine clearance did not indicate a decline in excretory renal function and were not significantly increased in anti-Fx1A-treated rats as compared to controls. The authors wished to provide information on SDMA in a glomerular toxicity model that complements work we have done examining the same marker in a renal tubular toxicity model (Evaluation of Renal Biomarkers, Including Symmetric Dimethylarginine, following Gentamicin-Induced Proximal Tubular Injury in the Rat. https://doi.org/10.34067/KID.0006542020). Our goal was to add to the body of literature to encourage investigators to consider alternative biomarkers to evaluate kidney function and injury. 

Other comments:

Lines 21-23. The authors state that SDMA correlates highly with GFR across multiple species including rats, dogs and humans. Line 69-71 this is repeated including cats. Based on current literature, this statement should be nuanced. Indeed, in dogs, Pelander et al. (2019) showed that overall diagnostic performance of SDMA as a marker of impaired GFR was the same as that of creatinine (both R2 = 0.62, P < .001) and this result was in line with McKenna et al. (2019) who concluded serum creatinine and SDMA were both only moderately correlated with the renal clearance (R2 = 0.52 and 0.27 respectively, p < .001). In cats, Brans et al. (2020) showed that the correlation between SDMA and GFR (τ B = −0.57; P < .001) was moderate and the correlation between sCr and GFR was of the same magnitude (τ B = −0.56; P < .001). Braff et al. (2014) equally observed that the relationship between SDMA and GFR (R2 = 0.82, P < .001) was very similar to the relationship between serum creatinine and GFR (R2 = 0.81, P <.001). Also in humans, a large meta-analysis of Kielstein et al. (2006) could not demonstrate outstanding correlation of 1/SDMA with GFR (r = 0.77 - 0.85, P < .001).

We thank the reviewer for this comment and wish to address these studies. While all three studies do address GFR and correlation to SDMA there are specific nuances to the populations used which weaken the context for the relationship between GFR measured and SDMA values ascertained and therefore should be regarded with some critical interpretation and not as complete proof of the inequity of SDMA to measure indirect GFR accurately. In Mckenna et al. 2019, samples used for GFR measurement were convenience samples sent to a reference laboratory for iohexol clearance. Inherently samples being sent for iohexol clearance testing usually infers concern for concentrating ability or kidney dysfunction. Non-CKD status would be difficult to discern in the population used as “healthy”. No independent assessment was used to confirm non-CKD or CKD status. The method for comparing disease state (GFR % decrease to mean GFR) is faulted due to the inability to confirm the “healthy population” is truly non-CKD in nature. A single time point is used for comparison of % GFR decrease to SDMA and creatinine, both in a disease state and “without disease.” GFR is highly variable even in healthy animals and multiple times points would be needed to accurately determine % loss in healthy or diseased dogs. Additionally, this study examined a disease state that requires multiple time points with persistent findings for appropriate diagnosis. In Pelander et al. 2019, there was significant overlap between healthy, inconclusive, and CKD Stage I dogs. Classification into one of these three groups appears slightly arbitrary and may introduce bias. CKD Stage I makes up ~50% of population used for analysis (N=26) vs CKD II (N=12), III (N=12), and IV (N=3). If subjective criteria for classification of Normal v CKD I dogs is used, this may introduce bias from the investigator. For the dogs reported as False Positives, there are no GFRs reported for interpretation. All of these dogs were classified as CKD I vs CKD II. As these categories appear slightly arbitrary this may bias the SDMA results and its ability to interpret disease. Given the false positives occurred with both creatinine and SDMA it is clear that animals with early changes to GFR should be assessed with multiple parameters which is already part of IDEXX Laboratories’ diagnostics suggestion. The sensitivity and specificity calculation for creatinine are concerning. The reported sensitivity of 90% is well above that which is generally accepted for creatinine in the diagnosis of early renal disease. This is likely due to calculating creatinine on the entire CKD population as an identifier of disease instead of an early indicator of disease in Stage 1 or suspected patients. With this same calculation applied to SDMA, the same concerns arise. In Brans et al. 2020, a very small sample size was used: cats with CKD (N=17), control cats (N=17), including a very limited borderline CKD population (N=5). The diabetic population included has no specific value to the evaluation of the CKD population. The clinical suggestion that a more sensitive and specific cutoff for SDMA might be 18 µg/dL is made on the limited sample above, and on “false positives” which did not have follow up to confirm SDMA levels did or did not remain persistent (i.e., SDMA was an earlier indicator). GFR methodology used is one published by the author, with limited evaluation and evidence in previous publications. A variance rate of 20-30%, and single sample time point testing was used for all the statistical comparisons in a CKD and control population. SDMA performs consistently, and in a similar fashion to sCr, supporting a good correlation to GFR in most cases and in combination with other kidney biomarkers (not a clinical vacuum) can provide valuable information on decline of kidney function in dogs and cats. This same philosophy can be applied to preclinical work – SDMA can provide additional information regarding GFR, specifically in populations where Cr is less than ideal due to muscle mass effect. The authors agree that “excellent” might suggest SDMA GFR correlation always exceeds creatinine’s correlation therefore it is reasonable to modify this to consistently correlates with GFR. We have made the change to line 22 in the manuscript.

Lines 139-150. Did the laboratory animal veterinarian use a scoring system for assessment of pain and well-being of the rats? If yes, please state so.

Based upon documents that described anti-Fx1A-treated rats and pre-study dose ranging investigations, no study-related pain, morbidity, or mortality was anticipated for rats in this study. Therefore, there were no study- specific pain criteria outside of the institution’s standard criteria. The contract research organization’s standard criteria for visible indicators of pain in rats on study included orbital tightening, nose/cheek flattening, changes in ear and whisker carriage, hunched posture, piloerection, and porphyrin staining around nose and muzzle. We have made changes to the text and have included the following references:

Sotocina SG, Sorge RE, Zaloum A, Tuttle AH, Martin LJ, Wieskopf JS, Mapplebeck JCS, Wei P, Zhan S, Zhang S, McDougall JJ, King OD, Mogil JS. The rat grimace scale: A partially automated method for quantifying pain in the laboratory rat via facial expressions Mol Pain 7: 55, 2011.

Akintola T, Raver C, Studlack P, Uddin O, Masri R, Keller A. The grimace scale reliably assesses chronic pain in a rodent model of trigeminal neuropathic pain. Neurobiology of Pain Vol 2, August 2017, Pages 13-17.

Lines 193-197. Please provide for all assays used: either appropriate references that include validation data or include validation data.

All assays used in this study were commercially available, validated assays. 

Urine cystatin C, µALB and NGAL levels were determined using the MILLIPLEX Rat Kidney Toxicity Magnetic Bead Panel 2 (RKTX2MAG-37K) EMD Millipore Corporation, Billerica, MA USA. Validation data can be found at the website:

https://www.emdmillipore.com/US/en/product/MILLIPLEX-MAP-Rat-Kidney-Toxicity-Magnetic-Bead-Panel-2-Toxicity-Multiplex-Assay,MM_NF-RKTX2MAG-37K

Urine clusterin and serum cystatin C levels were determined using Biovendor CLU (RD391034200CS) and Biovendor cystatin C (RD391009200R), BioVendor, Brno, Czech Republic. Biovendor CLU (RD391034200CS) is no longer on market, manufacturer has sent Analytical Performance pdf

. 

Validation data for serum cystatin C can be found at the website:

https://www.biovendor.com/cystatin-c-rat-elisa?d=114#technical-data

Urine KIM-1 and urine OPN were performed using R&D Systems ELISAs (RKM100 for KIM-1 and MOST00 for OPN) (R&D Systems, Inc. Minneapolis, MN USA). Validation data for OPN can be found at the website:

https://www.rndsystems.com/products/mouse-rat-osteopontin-opn-quantikine-elisa-kit_most00

Validation data for KIM-1 can be found at the website:

https://www.rndsystems.com/products/rat-tim-1-kim-1-havcr-quantikine-elisa-kit_rkm100#product-details

Serum SDMA was measured using the IDEXX SDMA® Test (IDEXX Laboratories, Inc, Westbrook, ME, USA). Validation data for KIM-1 can be found in the publication:

Evaluation of Renal Biomarkers, Including Symmetric Dimethylarginine, following Gentamicin-Induced Proximal Tubular Injury in the Rat. https://doi.org/10.34067/KID.0006542020

We have referenced the tests except for urine clusterin since the information is no longer available on the web and were uncertain how it should be referenced.

Lines 200-208. Especially if animals are sacrificed, the maximum information should be gathered from the samples/study. It is not unexpected that glomerular changes on light microscopy were minimal in the antiFx1A treated rats. It is unclear why ME or IF techniques were not used. This would have allowed to bring the interpretation of the renal lesions to another level.

The microscopic alterations in kidneys of rat models of human membranous nephritis have been reported in detail numerous times (Baker et al., 1989; Barabas et al., 2004 ab; Jefferson et al., 2010; Natori et al., 1987; Spicer et al., 2007). These reports documented the light microscopic changes, electron microscopic features, and unique alterations detectable by immunohistochemistry and Immunofluorescence. The focus of our investigation was to document and monitor changes in renal injury and function biomarkers, including SMDA, longitudinally in a rat passive Heymann nephritis model. We followed anti-Rat Fx1A serum dose and administration protocols for rats similar to those of several articles published in peer-reviewed journals (Natori et al., 1987; Jefferson et al., 2010). We included histologic examination of rat kidney at four times including conclusion of the study to document lesion development over time and consistency of lesions with previous studies. The study was conducted at a well-respected contract research organization and the gross necropsy and microscopic examination (primary and peer review) were conducted by board-certified veterinary pathologists. Our results are consistent with previous studies of experimentally induced Heymann nephritis in rats but were never intended to be the primary focus of the manuscript. Additional studies using electron microscopy or immunofluorescence were not aligned with our primary focus for this biomarker investigation. 

References:

• Baker PJ, et al. Depletion of C6 prevents development of proteinuria in experimental membranous nephropathy in rats. Am J Pathol 135(1): 185-194, 1989.

• Barabas AZ, et al. Presence of immunoglobulin M antibodies around the glomerular capillaries and in the mesangium of normal and passive Heymann nephritis rats. Int J Exp Path 85: 201-212, 2004a.

• Barabas AZ, et al., Production of Heymann nephritis by a chemically modified renal antigen. Int J Exp Path 85: 277-285, 2004b.

• Jefferson JA, et al. Experimental models of membranous nephropathy. Drug Discov Today Dis Model 7(1-2): 27-33, 2010.

• Natori Y, et al. Heymann nephritis in rats induced by human renal tubular antigens: characterization of antigen and antibody specificities. Clin Exp Immunol 69; 33-40, 1987.

• Spicer TS, et al. Induction of passive Heymann nephritis in complement component 6-deficient PVG rats. J Immunol 179(1): 172-178, 2007.

In the results section, the authors make too much subjective interpretation of their data (the terms minimal, slightly … are used numerous numerous times).

Changes have been made as requested.

I don’t see a good argument why statistics were not performed for proteinuria?

We thank the reviewer for the comment. We have conducted an analysis of the urine analytes measured by dip stick – protein, glucose, ketones, bilirubin, and blood. Only urinary protein was significantly increased in magnitude and frequency at all time points in anti-Fx1A-treated rats (P < 0.001). We have added this information to the manuscript.

Analyte Day p Value

Protein Day 9 <0.001

Protein Day 16 <0.001

Protein Day 21 <0.001

Protein Day 28 <0.001

Glucose Day 9 0.359

Glucose Day 16 NA†

Glucose Day 21 NA†

Glucose Day 28 NA†

Ketones Day 9 1.000

Ketones Day 16 0.713

Ketones Day 21 0.135

Ketones Day 28 0.013

Bilirubin Day 9 0.304

Bilirubin Day 16 1.000

Bilirubin Day 21 0.166

Bilirubin Day 28 0.596

Blood Day 9 0.102

Blood Day 16 0.359

Blood Day 21 1.000

Blood Day 28 0.002

†Insufficient variation to estimate p Value; all results from both groups were the same value.

Discussion: the first paragraph is too long (Lines 348-390).

We have edited and shortened the paragraph as requested.

Line 438-440: please provide references for this statement on cystatin C.

We have provided references as requested.

---

## [Editor Report · Decision Letter 1]

16 May 2022

Evaluation of renal injury and function biomarkers, including symmetric dimethylarginine (SDMA), in the rat passive Heymann nephritis (PHN) model

PONE-D-22-03958R1

Dear Dr. Michael J Coyne,

We’re pleased to inform you that your manuscript has been judged scientifically suitable for publication and will be formally accepted for publication once it meets all outstanding technical requirements.

Kind regards,

Prabhakar Orsu, PhD

Academic Editor

PLOS ONE

Additional Editor Comments (optional):

Accepted for Publication
---

## [Editor Report · Acceptance letter]

19 May 2022

PONE-D-22-03958R1 

Evaluation of renal injury and function biomarkers, including symmetric dimethylarginine (SDMA), in the rat passive Heymann nephritis (PHN) model 

Dear Dr. Coyne:

I'm pleased to inform you that your manuscript has been deemed suitable for publication in PLOS ONE. Congratulations! Your manuscript is now with our production department. 

Kind regards, 

on behalf of

Dr. Prabhakar Orsu 

Academic Editor

PLOS ONE